# THE HIGH-DIMENSIONAL GEOMETRY OF BINARY NEURAL NETWORKS

**Alexander G. Anderson**
Redwood Center for Theoretical Neuroscience
University of California, Berkeley
aga@berkeley.edu

**Cory P. Berg**
Redwood Center for Theoretical Neuroscience
University of California, Berkeley
cberg500@berkeley.edu

## ABSTRACT

Recent research has shown that one can train a neural network with binary weights and activations at train time by augmenting the weights with a high-precision continuous latent variable that accumulates small changes from stochastic gradient descent. However, there is a dearth of work to explain why one can effectively capture the features in data with binary weights and activations. Our main result is that the neural networks with binary weights and activations trained using the method of Courbariaux, Hubara et al. (2016) work because of the high-dimensional geometry of binary vectors. In particular, the ideal continuous vectors that extract out features in the intermediate representations of these BNNs are well-approximated by binary vectors in the sense that dot products are approximately preserved. Furthermore, the results and analysis used on BNNs are shown to generalize to neural networks with ternary weights and activations. Compared to previous research that demonstrated good classification performance with BNNs, our work explains why these BNNs work in terms of HD geometry. Our theory serves as a foundation for understanding not only BNNs but a variety of methods that seek to compress traditional neural networks. Furthermore, a better understanding of multilayer binary neural networks serves as a starting point for generalizing BNNs to other neural network architectures such as recurrent neural networks.

## 1 INTRODUCTION

The rapidly decreasing cost of computation has driven many successes in the field of deep learning in recent years. Consequently, researchers are now considering applications of deep learning in resource limited hardware such as neuromorphic chips, embedded devices and smart phones (Esser et al. (2016), Neftci et al. (2016), Andri et al. (2017)). A recent realization for both theoretical researchers and industry practitioners is that traditional neural networks can be compressed because they are highly over-parameterized. While there has been a large amount of experimental work dedicated to compressing neural networks (Sec. 2), we focus on the particular approach that replaces costly 32-bit floating point multiplications with cheap binary operations. Our analysis reveals a simple geometric picture based on the geometry of high dimensional binary vectors that allows us to understand the successes of the recent efforts to compress neural networks.

Courbariaux et al. (2016) and Hubara et al. (2016) showed that one can efficiently train neural networks with binary weights and activations that have similar performance to their continuous counterparts. Such BNNs execute 7 times faster using a dedicated GPU kernel at test time. Furthermore, they argue that such BNNs require at least a factor of 32 fewer memory accesses at test time that should result in an even larger energy savings. There are two key ideas in their papers (Fig. 1). First, a continuous weight, $w^c$, is associated with each binary weight, $w^b$, that accumulates small changes from stochastic gradient descent. Second, the non-differentiable binarize function ($\theta(x) = 1$ if $x > 0$ and $-1$ otherwise) is replaced with a continuous one during backpropagation. These modifications allow one to train neural networks that have binary weights and activations with stochastic gradient descent. While the work showed how to train such networks, the existence of neural networks with binary weights and activations needs to be reconciled with previous work that has sought to understand weight matrices as extracting out continuous features in data (e.g. Zeiler & Fergus (2014)). Summary of contributions:

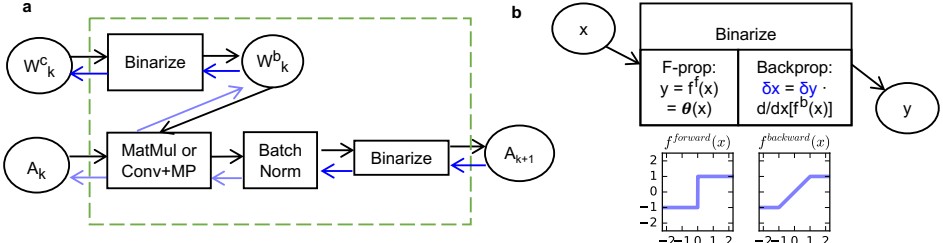

Figure 1: Review of the Courbariaux et al. (2016) BNN Training Algorithm: (a) A binary neural network is composed of binary convolution transformers (dashed green box). Each oval corresponds to a tensor and the derivative of the cost with respect to that tensor. Rectangles correspond to transformers that specify forward and backward propagation functions. Associated with each binary weight, $w^b$, is a continuous weight, $w^c$, that is used to accumulate gradients. $k$ denotes the $k$th layer of the network. (b) Each binarize transformer has a forward function and a backward function. The forward function simply binarizes the inputs. In the backward propagation step, one normally computes the derivative of the cost with respect to the input of a transformer via the Jacobian of the forward function and the derivative of the cost with respect to the output of that transformer ($\delta u \equiv dC/du$ where $C$ is the cost function used to train the network). Since the binarize function is non-differentiable, the straight-through estimator (Bengio et al. (2013)), which is a smoothed version of the forward function, is used for the backward function .

1. Angle Preservation Property: We demonstrate that binarization approximately preserves the direction of high-dimensional vectors. In particular, we show that the angle between a random vector (from a standard normal distribution) and its binarized version converges to $\arccos \sqrt{2/\pi} \approx 37°$ as the dimension of the vector goes to infinity. This angle is an exceedingly small angle in high dimensions. Furthermore, we show that this property is present in the weight vectors of a network trained using the method of Courbariaux et al. (2016).

2. Dot Product Proportionality Property: First, we empirically show that the weight-activation dot products, an important intermediate quantity a neural network, are approximately proportional under the binarization of the weight vectors. Next, we argue that if these weight activation dot products are proportional, then the continuous weights in the Courbariaux et al. (2016) method aren't just a learning artifact. The continuous weights obtained from the BNN training algorithm (which decouples the forward and backward propagation steps) are an approximation of the weights one would learn if the network were trained with continuous weights and regular backpropagation.

3. Generalized Binarization Transformation (GBT): We show that the computations done by the first layer of the network are fundamentally different than the computations being done in the rest of the network because correlations in the data result in high variance principal components that are not randomly oriented relative to the binarization. Thus we recommend an architecture that uses a continuous convolution for the first layer to embed the image in a high-dimensional binary space, after which it can be manipulated with cheap binary operations. Furthermore, we illustrate how a GBT (rotate, binarize, rotate back) is useful for embedding low dimensional data in a high-dimensional binary space.

4. Generalization to Ternary Neural Networks: We show that the same analysis applies to ternary neural networks. In particular, the angle between a random vector from a standard normal distribution and the ternarized version of that vector predicts the empirical distribution of such angles in a network trained on CIFAR10. Furthermore, the dot product proportionality property is shown to hold for ternary neural networks.

## 2 RELATED WORK

Neural networks that achieve good performance on tasks such as IMAGENET object recognition are highly computationally intensive. For instance, AlexNet has 61 million parameters and executes 1.5 billion operations to classify one 224 by 224 image (30 thousand operations/pixel) (Rastegari et al. (2016)). Researchers have sought to reduce this computational cost for embedded applications using a number of different approaches.

The first approach is to try and compress a pre-trained network. Kim et al. (2015) uses a Tucker decomposition of the kernel tensor and fine tunes the network afterwards. Han et al. (2015b) train a network, prune low magnitude connections, and retrain. Han et al. (2015a) extend their previous work to additionally include a weight sharing quantization step and Huffman coding of the weights. More recently, Han et al. (2017) train a dense network, sparsify it, and then retrain a dense network with the pruned weights initialized to zero. Second, researchers have sought to train networks using either low precision floating point numbers or fixed point numbers, which allow for cheaper multiplications (Courbariaux et al. (2014), Gupta et al. (2015), Judd et al. (2015), Gysel et al. (2016), Lin et al. (2016), Lai et al. (2017)).

Third, one can train networks that have quantized weights and or activations. Bengio et al. (2013) looked at estimators for the gradient through a stochastic binary unit. Courbariaux et al. (2015) train networks with binary weights, and then later with binary weights and activations (Courbariaux et al. (2016)). Rastegari et al. (2016) replace a continuous weight matrix with a scalar times a binary matrix (and have a similar approximation for weight activation dot products). Kim & Smaragdis (2016) train a network with weights restricted in the range $-1$ to $1$ and then use a noisy backpropagation scheme train a network with binary weights and activations. Alemdar et al. (2016), Li et al. (2016) and Zhu et al. (2016) focus on networks with ternary weights. Further work seeks to quantize the weights and activations in neural networks to an arbitrary number of bits (Zhou et al. (2016), Hubara et al. (2016)). Zhou et al. (2017) use weights and activations that are zero or powers of two. Lin et al. (2015) and Zhou et al. (2016) quantize backpropagation in addition to the forward propagation.

Beyond merely seeking to compress neural networks, the analysis of the internal representations of neural networks is useful for understanding how to to compress them. Agrawal et al. (2014) found that feature magnitudes in higher layers do not matter (e.g. binarizing features barely changes classification performance). Merolla et al. (2016) analyze the robustness of neural network representations to a collection of different distortions. Dosovitskiy & Brox (2016) observe that binarizing features in intermediate layers of a CNN and then using backpropagation to find an image with those features leads to relatively little distortion of the image compared to dropping out features. These papers naturally lead into our work where we are seeking to better understand the representations in neural networks based on the geometry of high-dimensional binary vectors.

## 3 THEORY AND EXPERIMENTS

We investigate the internal representations of neural networks with binary weights and activations. A binary neural network is trained on CIFAR-10 (same learning algorithm and architecture as in Courbariaux et al. (2016)). Experiments on MNIST were carried out using both fully connected and convolutional networks and produced similar results. The CIFAR-10 convolutional neural network has six layers of convolutions, all of which have a 3 by 3 spatial kernel. The number of feature maps in each layer are 128, 128, 256, 256, 512, 512. After the second, fourth, and sixth convolutions, there is a 2 by 2 max pooling operation. Then there are two fully connected layers with $1024$ units each. Each layer has a batch norm layer in between. The experiments using ternary neural networks use the same network architecture. The dimensionality of the weight vectors in these networks (i.e. convolution converted to a matrix multiply) is the patch size ($= 3 * 3 = 9$) times the number of channels.

### 3.1 PRESERVATION OF DIRECTION DURING BINARIZATION

In this section, we analyze the angle distributions (i.e. geometry) of high-dimensional *binary* vectors. This is crucial for understanding binary neural networks because we can imagine that at each layer of a neural network, there are some ideal continuous weight vectors that extract out features. A binary

neural network approximates these ideal continuous vectors with a binary vectors. In low dimensions, binarization strongly impacts the direction of a vector. However, we argue that binarization does *not* substantially change the direction of a high-dimensional continuous vector. It is often the case that the geometric properties of high-dimensional vectors are counter-intuitive. For instance, one key idea in the hyperdimensional computing theory of Kanerva (2009) is that two random, high-dimensional vectors of dimension $d$ whose entries are chosen uniformly from the set $\{-1, 1\}$ are approximately orthogonal. The result follows from the central limit theorem because the cosine angle between two such random vectors is normally distributed with $\mu = 0$ and $\sigma \sim 1/\sqrt{d}$. Then $\cos\theta \approx 0$ implies that $\theta \approx \frac{\pi}{2}$. Building upon this work, we study the way in which binary vectors are distributed relative to continuous vectors. As binarizing a continuous vector gives the binary vector closest in angle to that continuous vector, we can get a sense of how binary vectors are distributed relative to continuous vectors in high dimensions by binarizing continuous vectors. The standard normal distribution, which serves as an informative null distribution because it is rotationally invariant, is used to generate random continuous vectors which are then binarized. This analysis gives a fundamental insight into understanding the recent success of binary neural networks. Binarizing a random continuous vector changes its direction by a small amount *relative to* the angle between two random vectors in moderately high dimensions (Fig. 2a). Binarization changes the direction of a vector by approximately $37°$ in high dimensions. This seems like a large change based on our low-dimensional intuition. Indeed, the angle between two randomly chosen vectors from a rotationally invariant distribution is *uniform* in two dimensions. However, two randomly chosen vectors are approximately *orthogonal* in high dimensions. Thus while it is common for two random vectors to have an angle less than $37°$ in low dimensions, it is exceedingly rare in high dimensions. Therefore $37°$ is a small angle in high dimensions.

In order to test our theory of the binarization of random vectors chosen from a rotationally invariant distribution, we train a multilayer binary CNN on CIFAR10 (using the Courbariaux et al. (2016) method) and study the weight vectors[1] of that network. Remarkably, there is a close correspondence between the experimental results and the theory for the angles between the binary and continuous weights (Fig. 2b). For each layer, the distribution of the angles between the binary and continuous weights is sharply peaked near the $d \to \infty$ expectation of $\arccos\sqrt{2/\pi}$. We note that there is a small but systematic deviation from the theory towards larger angles for the higher layers of the network (Fig. 6). Ternary neural networks are considered in (SI Sec. 5.5) and yield a similar result.

### 3.2 DOT PRODUCT PROPORTIONALITY AS A SUFFICIENT CONDITION FOR APPROXIMATING A NETWORK WITH CONTINUOUS WEIGHTS

Given the previous discussion, an important question to ask is: are the so-called continuous weights a learning artifact without a clear correspondence to the binary weights? While we know that $w^b = \theta(w^c)$, there are many continuous weights that map onto a particular binary weight vector. Which one is found when using the straight-through estimator to backpropagate through the binarize function? Remarkably, there is a clear answer to this question. In numerical experiments, we see that one gets the continuous weight vector such that the dot products of the activations with the pre-binarization and post-binarization weights are highly correlated (Fig. 3). In equations, $a \cdot w^b \sim a \cdot w^c$. We call this relation the Dot Product Proportionality (DPP) property. The proportionality constant, which is subsequently normalized away by a batch norm layer, depends on the magnitudes of the continuous and binary weight vectors and the cosine angle between the binary and continuous weight vectors. The theoretical consequences of the DPP property are explored in the rest of this section.

We show that the modified gradient of the BNN training algorithm can be viewed as an estimator of the gradient that would be used to train the continuous weights in traditional backpropagation. This establishes the fundamental point that while the weights and activations are technically binary, they are operating as if the weights are continuous. For instance, one could imagine using an exhaustive search over all binary weights in the network. However, the additional structure in the problem associated with taking dot products makes the optimization simpler than that. Furthermore, we show that if the dot products of the activations with the pre-binarized and post-binarized weights are proportional then straight-through estimator gradient is proportional to the continuous weight network gradient. The key to the analysis is to focus on the transformers in the network whose forward and

---

[1]If each convolution is written as the matrix multiplication $Wx$ where $x$ is a column vector, then the weight vectors are the rows of $W$.

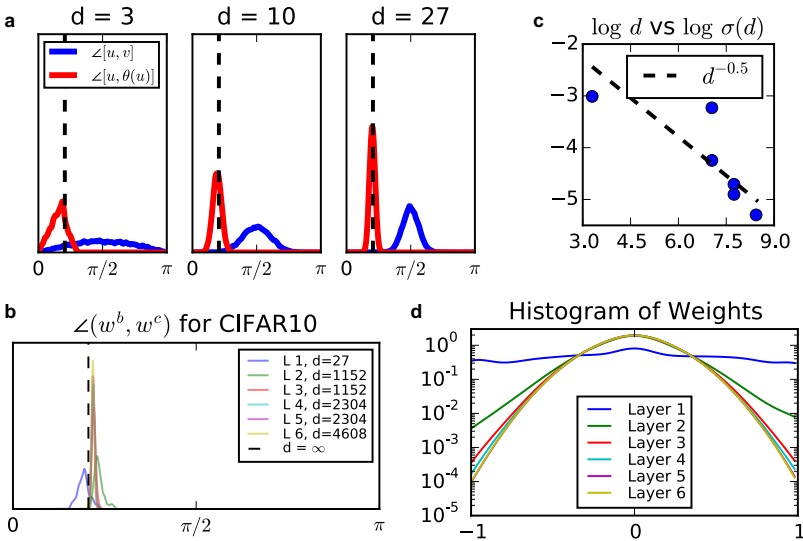

Figure 2: Binarization of High-Dimensional Vectors Approximately Preserves their Direction in Theory and Practice: (a) Distribution of angles between two random vectors (blue), and between a vector and its binarized version (red), for a rotationally invariant distribution of dimension $d$. The red distribution is peaked near the $d \to \infty$ limit of $\arccos \sqrt{2/\pi} \approx 37°$ (SI, Sec. 1). While $37°$ may seem like a large angle, that angle is small as compared to the angle between two random vectors in moderately high dimensions (i.e. the blue and red curves are well-separated). (b) Angle distribution between continuous and binary weight vectors by layer for a binary CNN trained on CIFAR-10. For the higher layers, there is a close correspondence to the theory. There is a small, but systematic deviation towards large angles (SI, Fig. 6). $d$ is the dimension of the filters at each layer. (c) Standard deviations of the angle distributions from (b) by layer. We see a correspondence to the theoretical expectation that standard deviations of each of the angle distributions scales as $d^{-0.5}$ (SI, Sec. 1). (d) Histogram of the components of the continuous weights at each layer. The distribution is approximately Gaussian for all but the first layer. Furthermore, there is a high density of weights near zero, which is the threshold for the binarization function.

backward propagation functions are not related in the way that they would normally be related in typical gradient descent.

Suppose that there is a neural network where two tensors, $u$, and $v$ and the associated derivatives of the cost with respect to those tensors, $\delta u$, and $\delta v$, are allocated. Suppose that the loss as a function of $v$ is $L(x)|_{x=v}$. Further, suppose that there are two potential forward propagation functions, $f$, and $g$. If the network is trained under normal conditions using $g$ as the forward propagation function, then the following computations are done:

$$v \leftarrow g(u) \qquad \delta v \leftarrow L'(x = v = g(u)) \qquad \delta u \leftarrow \delta v \cdot g'(u)$$

where $L'(x)$ denotes the derivative of $L$ with respect to the vector $x$. In a modified backpropagation scheme, the following computations are done

$$v \leftarrow f(u) \qquad \delta v = L'(x = v = f(u)) \qquad \delta u \leftarrow \delta v \cdot g'(u)$$

A sufficient condition for $\delta u$ to be the same in both cases is $L'(x = f(u)) \sim L'(x = g(u))$ where $a \sim b$ means that the vector $a$ is a scalar times the vector $b$.

Now this general observation is applied to the binarize transformer of Fig. 1. Here, $u$ is the continuous weight, $w^c$, $f(u)$ is the pointwise binarize function, $g(u)$ is the identity function[2], and $L$ is the loss of the network as a function of the weights in a particular layer. Given the network architecture, $L(x) = M(a \cdot x)$ where $a$ are the activations corresponding to that layer and $M$ is the loss as a

---

[2]For the weights, $g$ as in Fig. 1 is the identity function because the $w^c$'s are clipped to be in the range $[-1, 1]$.

function of the weight-activation dot products. Then $L'(x) = M'(a \cdot x) \odot a$ where $\odot$ denotes a pointwise multiply. Thus the sufficient condition is $M'(a \cdot w^b) \sim M'(a \cdot w^c)$. Since the dot products are followed by a batch normalization, $M(k\vec{x}) = M(\vec{x}) \rightarrow M'(\vec{x}) = kM'(k\vec{x})$. Therefore, it is sufficient that $a \cdot w^b \sim a \cdot w^c$, which is the DPP property. When the DPP only approximately holds, the second derivative can be used to bound the error between the two gradients of the two learning procedures. In summary, the learning dynamics where $g$ is used for the forward and backward passes (i.e. training the network with continuous weights) is approximately equivalent to the modified learning dynamics ($f$ on the forward pass, and $g$ on the backward pass) when we have the DPP property.

While we demonstrated that the BNN learning dynamics approximate the dynamics that one would have by training a network with continuous weights using a mixture of empirical and theoretical arguments, the ideal result would be that the learning algorithm implies the DPP property. It should be noted that in the case of stochastic binarization where $E(w^b) = w^c$ is chosen by definition, the DPP property is true by design. However, it is remarkable that the property still holds in the case of deterministic binarization, which is revealing of the fundamental nature of the representations used in neural networks.

While the main focus of this section is the binarization of the weights, the arguments presented can also be applied to the binarize block that corresponds to the non-linearity of the network. The analogue of the DPP property for this binarize block is: $w^b \cdot a^c \sim w^b \cdot a^b$ where $a^c$ denotes the pre-binarized (post-batch norm) activations and $a^b = a$ denotes the binarized activations. This property is empirically verified to hold. For the sake of completeness, the dot product histogram corresponding to $w^c \cdot a^c \sim w^b \cdot a^b$ is also computed, although it doesn't directly correspond to removing one instance of a binarize transformer. This property is also empirically verified to hold (SI, Fig. 5).

Impact on Classification: It is natural to ask to what extent the classification performance depends on the binarization of the weights. In experiments on CIFAR10, if the binarization of the weights on all of the convolutional layers is removed, the classification performance drops by only 3 percent relative to the original network. Looking at each layer individually, removing the weight binarization for the first layer accounts for this entire percentage, and removing the binarization of the weights for each other layer causes no degradation in performance. This result is evident by looking at the 2D dot product histograms in Fig 3. The off-diagonal quadrants show where switching the weights from binary to continuous changes the sign of the binarized weight-activation dot product. In all of the layers except the first layer, there are very few dot products in the off-diagonal quadrants. Thus we recommend the use of the dot product histograms for studying the performance of binary neural networks. Removing the binarization of the activations has a substantial impact on the classification performance because that removes the main non-linearity of the network.

### 3.3    INPUT CORRELATIONS AND THE GENERALIZED BINARIZATION TRANSFORMATION

Not surprisingly, some distributions are impacted more strongly by binarization than others. A binary neural network must adapt its internal representations in such a way to not be degraded too much by binarization at each layer. In this section we explore the idea that the principal components of the input to the binarization function should be randomly oriented relative to the binarization. While the network can adapt the higher level representations to satisfy this property, the part of the network that interfaces with the input doesn't have that flexibility. We make the novel observation that the difficulties in training the first layer of the network are tied to the intrinsic correlations in the input data. In order to be more precise, we define the Generalized Binarization Transformation (GBT)

$$\theta_R(x) = R^T \theta(Rx)$$

where $x$ is a column vector, $R$ is a fixed rotation matrix, and $\theta$ is the pointwise binarization function from before. The rows of $R$ are called the axes of binarization. If $R$ is the identity matrix, then $\theta_R = \theta$ and the axes of binarization are the canonical basis vectors $(..., 0, 1, 0, ...)$. $R$ can either be chosen strategically or randomly.

The GBT changes the distribution being binarized through a rotation. For appropriate choices of the rotation, $R$, the directions of the input vectors, $x$, are changed insignificantly by binarization. The angle between a vector and its binarized version is dependent on the dot product: $x \cdot \theta_R(x)$, which is equal to $x^T \theta_R(x) = (Rx)^T \theta(Rx) = y \cdot \theta(y)$ where $y = Rx$. As a concrete example of the benefits

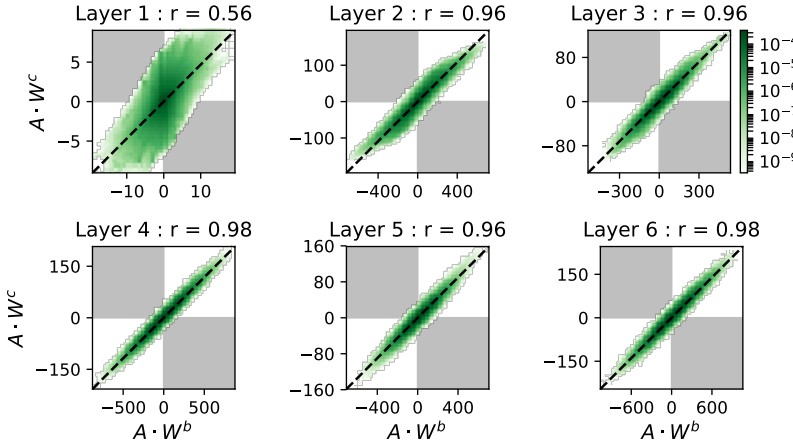

Figure 3: Binarization Preserves Dot Products: Each subplot shows a 2D histogram of the dot products between the binarized weights and the activations (horizontal axis) and the dot products between the continuous weights and the activations (vertical axis) for different layers of a network trained on CIFAR10. Surprisingly, the dot products are *highly* correlated ($r$ is the Pearson correlation coefficient). Thus replacing $w^b$ with $w^c$ changes the overall constant in front of the dot products, while still preserving whether the dot product is zero or not zero. This overall constant is divided out by the subsequent batch norm layer. The shaded quadrants correspond to dot products where the sign changes when replacing the binary weights with the continuous weights. Notice that for all but the first layer, a very small fraction of the dot products lie in these off diagonal quadrants. The top left figure (labeled as Layer 1) corresponds to the input and the first convolution. Note that the correlation is weaker in the first layer.

of the GBT, consider the case where $x \sim N(0, \Sigma)$ and $\Sigma_{i,j} = \delta_{i,j} \exp(2ki)$ for $k = 0.1$ (therefore $y \sim N(0, R\Sigma R^T)$). As the dimension goes to infinity, the angle between a vector drawn from this distribution and its binarized version approaches $\pi/2$. Thus binarization is destructive to vectors from this distribution. However, if the GBT is applied with a fixed random matrix[3], the angle between the vector and its binarized version converges to $37°$ (Fig. 4). Thus a random rotation can compensate for the errors incurred from directly binarizing a non-isotropic Gaussian.

Moving into how this analysis applies to a binary neural network, the network weights must approximate the important directions in the activations using binary vectors. For instance, Gabor filters are intrinsic features in natural images and are often found in the first layer weights of neural networks trained on natural images (e.g. Olshausen et al. (1996); Krizhevsky et al. (2012)). While the network has flexibility in the higher layers, the first layer must interface directly with the input where the features are not necessarily randomly oriented. For instance, consider the 27 dimensional input to the first set of convolutions in our network: 3 color channels of a 3 by 3 patch of an image from CIFAR10 with the mean removed. 3 PCs capture 90 percent of the variance of this data and 4 PCs capture 94.5 percent of the variance. The first two PCs are spatially uniform colors. More generally, large images such as those in IMAGENET have the same issue. Translation invariance of the image covariance matrix implies that the principal components are the filters of the 2D Fourier transform. Scale invariance implies a $1/f^2$ power spectrum, which results in the largest PCs corresponding to low frequencies (Field (1987)).

Another manifestation of this issue can be seen in our trained networks. The first layer has a much smaller dot product correlation than the other layers (Fig. 3). To study this, we randomly permute the activations in order to generate a distribution with the same marginal statistics as the original data but independent joint statistics (a different permutation for each input image). Such a transformation gives a distribution with a correlation equal to the normalized dot product of the weight vectors

---

[3]Random rotation matrices are chosen from the Haar distribution on SO(3) using the method of Stewart (1980).

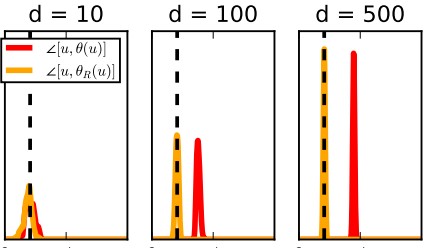 

Figure 4: Left: Random Rotation Improves Angle Preservation for a Non-Isotropic Gaussian. Random vectors are drawn from a Gaussian of dimension $d$ with a diagonal covariance matrix whose entries vary exponentially. As in Fig. 2, the red curve shows the angle between a random vector and its binarized version. Since the Gaussian is no longer isotropic, the red curve no longer peaks at $\theta = \arccos \sqrt{2/\pi}$. However, if the binarization is replaced with a GBT with a fixed random matrix, the direction of the vector is again approximately preserved. Right: Permuting the activations shows that the correlations observed in Fig. 3 are not merely due to correlations between the binary and continuous weight vectors. The correlations are due to these weight vectors corresponding to high variance directions in the data.

(SI Sec. 3). The correlations for the higher layers decrease substantially but the correlation in the first layer *increases* (Fig. 4, right). For the first layer, the shuffling operation randomly permutes the pixels in the image. Thus we demonstrate that the binary weight vectors in the first layer are not well-aligned with the continuous weight vectors relative to the input data. Our theoretically grounded analysis is consistent with previous work. Han et al. (2015b) find that compressing the first set of convolutional weights of a particular layer by the same fraction has the highest impact on performance if done on the first layer. Zhou et al. (2016) find that accuracy degrades by about 0.5 to 1 percent on SHVN when quantizing the first layer weights. Thus it is recommended to rotate the input data before normalization or to use continuous weights for the first layer.

## 4    CONCLUSION

Neural networks with binary weights and activations have similar performance to their continuous counterparts with substantially reduced execution time and power usage. We provide an experimentally verified theory for understanding how one can get away with such a massive reduction in precision based on the geometry of HD vectors. First, we show that binarization of high-dimensional vectors preserves their direction in the sense that the angle between a random vector and its binarized version is much smaller than the angle between two random vectors (Angle Preservation Property). Second, we take the perspective of the network and show that binarization approximately preserves weight-activation dot products (Dot Product Proportionality Property). More generally, when using a network compression technique, we recommend looking at the weight activation dot product histograms as a heuristic to help localize the layers that are most responsible for performance degradation. Third, we discuss the impacts of the low effective dimensionality of the data on the first layer of the network. We recommend either using continuous weights for the first layer or a Generalized Binarization Transformation. Such a transformation may be useful for architectures like LSTMs where the update for the hidden state declares a particular set of axes to be important (e.g. by taking the pointwise multiply of the forget gates with the cell state). Finally, we show that neural networks with ternary weights and activations can also be understood with our approach. More broadly speaking, our theory is useful for analyzing a variety of neural network compression techniques that transform the weights, activations or both to reduce the execution cost without degrading performance.

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

## 5 SUPPLEMENTARY INFORMATION

### 5.1 EXPECTED ANGLES

Random $n$ dimensional vectors are drawn from a rotationally invariant distribution. The angle between two random vectors and the angle between a vector and its binarized version are compared. A rotationally invariant distribution can be factorized into a pdf for the magnitude of the vector times a distribution on angles. In the expectations that we are calculating, the magnitude cancels out and there is only one rotationally invariant distribution on angles. Thus it suffices to compute these expectations using a Gaussian.

Lemmas:

1. Consider a vector, $v$, chosen from a standard normal distribution of dimension $n$. Let $\rho = \frac{v_1}{\sqrt{v_1^2 + \ldots + v_n^2}}$. Then $\rho$ is distributed according to: $g(\rho) = \frac{1}{\sqrt{\pi}} \frac{\Gamma(n/2)}{\Gamma((n-1)/2)} (1 - \rho^2)^{\frac{n-3}{2}}$ where $\Gamma$ is the Gamma function.

   Proof: Begin by considering the integral $G(\rho_0) = p(\rho < \rho_0) = \int \prod_i dv_i p(v) I(\rho(v) < \rho_0)$ where $I$ is an indicator function. The desired distribution comes from taking the derivative of this cumulative distribution $g(\rho_0) = \frac{d}{d\rho_0} G(\rho_0)$. Consider the generalized spherical coordinate transformation: $v_1 = r \cos \phi_1$, $v_2 = r \sin \phi_1 \cos \phi_2$, $v_3 = r \sin \phi_1 \sin \phi_2 \cos \phi_3$, ... $v_{n-1} = r \sin \phi_1 \cdots \sin \phi_{n-2} \cos \phi_{n-1}$, $v_n = r \sin \phi_1 \cdots \sin \phi_{n-2} \sin \phi_{n-1}$ where $\phi_1, \ldots, \phi_{n-2} \in [0, \pi]$ and $\phi_{n-1} \in [0, 2\pi]$. The generalized spherical volume element is $dv = r^{n-1} \sin^{n-2}(\phi_1) \sin^{n-3}(\phi_2) \cdots \sin(\phi_{n-2}) dr d\phi_1 \cdots d\phi_{n-1}$. [4] Thus we can write out the integral

   $$G(\rho_0) = \int I(\cos \phi_1 < \rho_0) \frac{e^{-r^2/2}}{\sqrt{(2\pi)^n}} r^{n-1} \sin^{n-2}(\phi_1) \sin^{n-3}(\phi_2) \cdots \sin(\phi_{n-2}) dr d\phi_1 \cdots d\phi_{n-1}$$

   The integral factorizes and all of the terms are independent of $\rho_0$ except the integral over $\phi_1 \equiv \phi$. Thus $G(\rho_0) \sim \int_0^\pi I(\cos \phi < \rho_0) \sin^{n-2}(\phi) d\phi$. Using the substitution $\rho = \cos \phi$ (which is also consistent with the definition of $\rho$ above), $d\rho = -\sin(\phi) d\phi$, $\sin \rho = (1 - \rho^2)^{0.5}$, so $G(\rho_0) \sim \int_{-1}^1 I(\rho < \rho_0)(1 - \rho^2)^{(n-3)/2} d\rho$. Taking the derivative with respect to $\rho_0$ and using the fundamental theorem of calculus gives $g(\rho) \sim (1 - \rho^2)^{(n-3)/2}$. The normalization constant is equal to a beta function that evaluates to the desired result (substitute $t = \rho^2$).

2. $\frac{\Gamma(z+\alpha)}{\Gamma(z+\beta)} = z^{\alpha-\beta} \left(1 + \frac{(\alpha-\beta)(\alpha+\beta+1)}{2z}\right) + O(|z|^{-2})$ as $z \to \infty$

Cases:

- Distribution of angles between two random vectors.

  Since a Gaussian is a rotationally invariant distribution, we can say without loss of generality that one of the vectors is $(1, 0, 0, \ldots 0)$. Then the cosine angle between those two vectors is $\rho$ as defined above. While the exact distribution of $\rho$ is given by Lemma 1, we note that

  - $E(\rho) = 0$ due to the symmetry of the distribution.
  - $Var(\rho) = E(\rho^2) = \frac{1}{n}$ because $1 = E\left(\frac{\sum_i x_i^2}{\sum_j x_j^2}\right) = \sum_i E\left(\frac{x_i^2}{\sum_j x_j^2}\right) = n * E(\rho^2)$

- Angles between a vector and the binarized version of that vector, $\eta = \frac{v \cdot \theta(v)}{||v|| \cdot ||\theta(v)||} = \frac{\sum_i |v_i|}{\sqrt{\sum v_i^2} * \sqrt{n}}$

  -
  $$E(\eta) = \frac{\sqrt{n}}{\sqrt{\pi}} * \frac{\Gamma(n/2)}{\Gamma((n+1)/2)} \qquad \lim_{n \to \infty} E(\eta) = \sqrt{\frac{2}{\pi}}$$

  First, we note $E(\eta) = \sqrt{n} E(|\rho|)$. Then $E(|\rho|) = \int_0^1 d\rho \, \rho \frac{2}{\sqrt{\pi}} \frac{\Gamma(n/2)}{\Gamma((n-1)/2)} (1 - \rho^2)^{\frac{n-3}{2}} = \frac{2}{\sqrt{\pi}} * \frac{1}{n-1} \frac{\Gamma(n/2)}{\Gamma((n-1)/2)}$ (substitute $u = \rho^2$ and use $\Gamma(x + 1) = x\Gamma(x)$ ). Lemma

---

[4]https://en.wikipedia.org/wiki/N-sphere#Spherical_coordinates

two gives the $n \to \infty$ limit. $\frac{2}{\sqrt{\pi}} \frac{\sqrt{n}}{n-1} \frac{\Gamma(n/2)}{\Gamma((n-1)/2)} \approx \frac{2}{\sqrt{\pi}} \frac{\sqrt{n}}{n-1} \sqrt{\frac{n}{2}} \left[1 + \frac{0.5*0.5}{2(n/2)}\right] = \sqrt{\frac{2}{\pi}} \left(1 + \frac{5}{4} * \frac{1}{n}\right) + O(1/n^2)$

$$Var(\eta) = \frac{1}{n}\left(1 - \frac{1}{\pi}\right) + O(1/n^2)$$

Thus we have the normal scaling as in the central limit theorem of the large $n$ variance. We can calculate this explicitly following the approach of [5].

As $E(\eta)$ has been calculated, it suffices to calculate $E(\eta^2)$. Expanding out $\eta^2$, $E(\eta^2) = \frac{1}{n} + (n-1) * E(\frac{|v_1 v_2|}{v_1^2 + ... v_n^2})$. Below we show that $E(\frac{|v_1 v_2|}{v_1^2 + ... v_n^2}) = \frac{2}{\pi n}$. Thus the variance is:

$$\frac{1}{n} * \left(1 - \frac{2}{\pi}\right) + \frac{2}{\pi} - \left(\frac{\sqrt{n}}{\sqrt{\pi}} \frac{\Gamma(n/2)}{\Gamma((n+1)/2)}\right)^2$$

Using Lemma 2 to expand out the last term, we get $[\frac{\sqrt{n}}{\sqrt{\pi}}(n/2)^{-1/2}(1 - 1/(4n) + O(n^{-2}))]^2 = \frac{2}{\pi}(1 - 1/(2n) + O(n^{-2}))$. Plugging this in gives the desired result.

Going back to the calculation of that expectation, change variables to $v_1 = r\cos\theta$, $v_2 = r\sin\theta$, $z^2 = v_3^2 + ... + v_n^2$. The integration over the volume element $dv_3 ... dv_n$ is rewritten as $dz dA_{n-3}$ where $dA_n$ denotes the surface element of a $n$ sphere. Since the integrand only depends on the magnitude, $z$, $\int dA_{n-3} = z^{n-3} * S_{n-3}$ where $S_n = \frac{2\pi^{(n+1)/2}}{\Gamma(\frac{n+1}{2})}$ denotes the surface area of a unit $n$-sphere. Then

$$E\left(\frac{|v_1 v_2|}{v_1^2 + ... v_n^2}\right) = (2\pi)^{-n/2} S_{n-3} \int_0^{2\pi} d\theta |\cos\theta\sin\theta| * \int r dr z^{n-3} dz * \frac{r^2}{r^2 + z^2} * e^{-(z^2 + r^2)/2}$$

Then substitute $r = p\cos\phi$, $z = p\sin\phi$ where $\phi \in [0, \pi/2]$

$$= (2\pi)^{-n/2} * 2S_{n-3} \int_0^{\pi/2} d\phi \cos\phi^3 * \sin\phi^{n-3} \int_0^\infty dp * p^{n-1} e^{-p^2/2}$$

The first integral is $\frac{2}{n(n-2)}$ using $u = \sin^2\phi$. The second integral is $2^{(n-2)/2}\Gamma(n/2)$ using $u = p^2/2$ and the definition of the gamma function. Simplifying, the result is $\frac{2}{\pi*n}$.

Thus the angle between a vector and a binarized version of that vector converges to $\arccos\sqrt{\frac{2}{\pi}} \approx 37°$, which is a very small angle in high dimensions.

## 5.2 AN EXPLICIT EXAMPLE OF LEARNING DYNAMICS

In this subsection, we look at the learning dynamics for the BNN training algorithm in a simple case and gain some insight about the learning algorithm. Consider the case of regression where the target output, $y$, is predicted with a binary linear predictor with $x$ as the input. Using a squared error loss, $L = (y - \hat{y})^2 = (y - w^b x)^2 = (y - \theta(w^c)x)^2$. (In this notation, $x$ is a column vector.) Taking the derivative of this loss with respect to the continuous weights and using the rule for back propagating through the binarize function gives $\Delta w^c \sim -dL/dw^c = -dL/dw^b \cdot dw^b/dw^c = (y - w^b x)x^T$. Finally, averaging over the training data gives

$$\Delta w^c \sim C_{yx} - \theta(w^c) \cdot C_{xx} \qquad C_{yx} = E[yx^T] \qquad C_{xx} = E(xx^T) \qquad (1)$$

It is worthwhile to compare this equation the corresponding equation from typical linear regression: $\Delta w^c \sim C_{yx} - w^c \cdot C_{xx}$. For simplicity, consider the case where $C_{xx}$ is the identity matrix. In this case, all of the components of $w$ become independent: $\delta w = \epsilon * (\alpha - \theta(w))$ where $\epsilon$ is the learning rate and $\alpha$ is the entry of $C_{yx}$ corresponding to a particular element, $w$. Compared to regular linear regression, it is clear that the stable point of these equations is when $w = \alpha$. Since the weight is

---

[5]https://en.wikipedia.org/wiki/Volume_of_an_n-ball#Gaussian_integrals

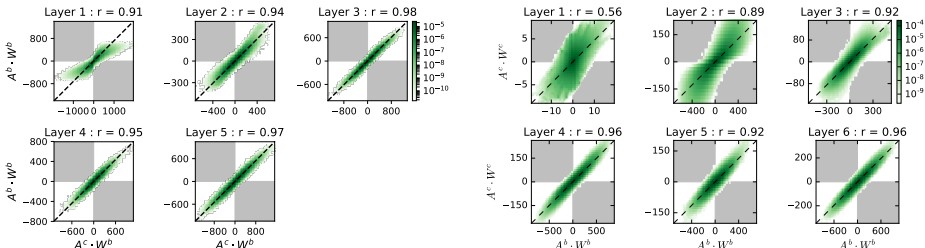

Figure 5: Activation Binarization Preserves Dot Products: Left: Each panel shows a 2D histogram of the dot products between the binarized weights and binarized activations (vertical axis) and post-batch norm (but pre-activation binarization) activations (horizontal axis). The binarization transformer does little to corrupt the dot products between weights and activations. Right: Dot products between the binary weights and binary activations (horizontal axis) compared against the dot products between the continuous weights and continuous activations (vertical axis). The dot products are not significantly impacted by removing binarization.

binarized, that equation cannot be satisfied. However, it can be shown ($\star$) that in this special case of binary weight linear regression, $E(\theta(w^c)) = \alpha$. Intuitively, if we consider a high dimensional vector and the fluctuations of each component are likely to be out of phase, then $w^b \cdot x \approx w^c \cdot x$ is going to be correct in expectation with a variance that scales as $\frac{1}{n}$. During the actual learning process, we anneal the learning rate to a very small number, so the particular state of a fluctuating component of the vector is frozen in. Relatedly, the equation $C_{yx} \approx w C_{xx}$ is easier to satisfy in high dimensions, whereas in low dimensions, it is only satisfied in expectation.

Proof for ($\star$): Suppose that $|\alpha| \leq 1$. The basic idea of these dynamics is that steps of size proportional to $\epsilon$ are taken whose direction depends on whether $w > 0$ or $w < 0$. In particular, if $w > 0$, then the step is $-\epsilon \cdot |1 - \alpha|$ and if $w < 0$, the step is $\epsilon \cdot (\alpha + 1)$. It is evident that after a sufficient burn-in period, $|w| \leq \epsilon * \max(|1 - \alpha|, 1 + \alpha) \leq 2\epsilon$. Suppose $w > 0$ occurs with fraction $p$ and $w < 0$ occurs with fraction $1 - p$. In order for $w$ to be in equilibrium, oscillating about zero, these steps balance out on average: $p(1 - \alpha) = (1 - p)(1 + \alpha) \rightarrow p = (1 + \alpha)/2$. Then the expected value of $\theta(w)$ is $1 * p + (-1) * (1 - p) = \alpha$. When $|\alpha| > 1$, the dynamics diverge because $\alpha - \theta(w)$ will always have the same sign. This divergence demonstrates the importance of some normalization technique such as batch normalization or attempting to represent $w$ with a constant times a binary matrix.

## 5.3 DOT PRODUCT CORRELATIONS AFTER ACTIVATION PERMUTATION

Suppose that $A = w \cdot a$ and $B = v \cdot a$ where $w, v$ are weight vectors and $a$ is the vector of activations. What is the correlation, $r$, between $A$ and $B$? Assuming that $E(a) = 0$, $E(A) = E(B) = 0$. Then

$E(AB) = \sum_{i,j} w_i v_j E(a_i a_j) = w^T C v$ where $C_{i,j} = E(a_i a_j)$. Setting $v = w$ and $w = v$ gives $E(A^2) = w^T C w$ and $v^T C v$. Thus $r = \frac{w^T C v}{\sqrt{(w^T C w)(v^T C v)}}$.

In the case where the activations are randomly permuted, $C$ is proportional to the identity matrix, and thus the correlation between $A$ and $B$ is the cosine angle between $u$ and $v$.

## 5.4 ANGLE PLOT

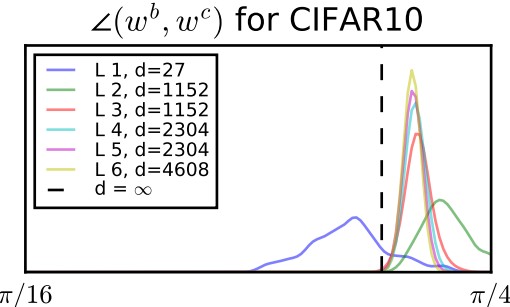

Figure 6: Angle distribution between continuous and binary weight vectors by layer for a binary CNN trained on CIFAR-10 (same plot as in Fig. 2b except zoomed in). Notice that there is a small but systematic deviation towards larger angles relative to the theoretical expectation (vertical dotted line). As the dimension of the vectors in the layer goes up, the distribution gets sharper. The theory predicts that the standard deviation of these distributions scales as $1/\sqrt{d}$. This relationship is shown to approximately hold in Fig. 2c.

## 5.5 TERNARY NEURAL NETWORKS

Moving beyond binarization, recent work has shown how to train neural networks where the activations are quantized to three (or more) values (e.g. Hubara et al. (2016)). Indeed, ternarization may be a more natural quantization method than binarization for neural network weights because one can express a positive association $(+1)$, a negative association $(-1)$, or no association $(0)$ between two features in a neural network. We show that the analysis used on BNNs holds for ternary neural networks.

The quantization function: $\text{ter}_a(x) = 1$ if $x > a$, $0$ if $|x| < a$, and $-1$ if $x < -a$ is used in place of the binarize function with the same straight-through estimator for the gradient. Call $a$ the ternarization threshold. The exact value of $a$ is only important at initialization because the scaling constant of the batch normalization layer allows the network to adapt the standard deviation of the pre-nonlinearity activations to the value of $a$. The network architecture from the previous experiments was used to classify images in CIFAR-10 and $a$ was chosen to be equal to $0.02$. In practice, roughly 10 percent of the ternarized weights were zero (Fig. 7a) and 2 percent of the activations were zero. Thus the network learning process did not ignore the possibility of using zero weights. However, more work is needed to effectively use the zero value for the activations.

The empirical distribution of angles between the continuous vectors and their ternarized counterparts is highly peaked at the value predicted by the theory (Fig. 7b). Random vectors are chosen from a standard normal distribution of dimension $d$. As in the case of binarization, the ternarized version of a vector is close in angle to the original vector in high dimensions (Fig. 7c). These vectors are quantized using $\text{ter}_a$ for different values of $a$. The peak angle varies substantially as a function of $a$ (Fig. 7d). Note that for $a = 0$, the ternarization function collapses into the binarization function. The empirical value of $a$ is the ratio of the empirical threshold to the empirical standard deviation of the continuous weights. Thus $a \approx 0.02/0.18 \approx 0.11$ for the higher layers. Remarkably, the theoretical prediction for the peak angle as a function of $a$ matches closely with the empirical result (Fig. 7b).

Finally, the dot product proportionality property is also shown to hold for ternary neural networks (Fig. 7e). Thus the continuous weights found using the TNN training algorithm approximate the continuous weights that one would get if the network were trained with continuous weights and regular backpropagation.

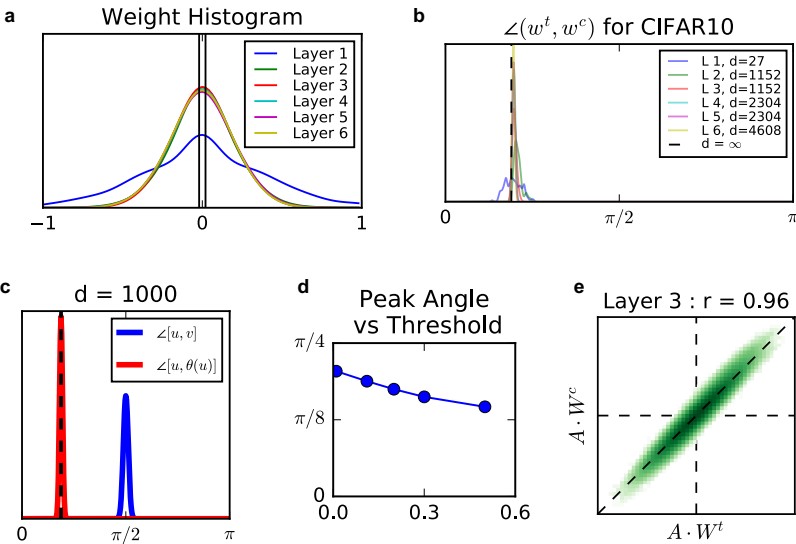

Figure 7: Ternarization of High-Dimensional Vectors Preserves their Direction in Theory and Practice: (a) Histogram of the components of the continuous weights at each layer for a ternary CNN trained on CIFAR-10. The distribution is approximately Gaussian for all but the first layer. The vertical lines show the ternarization thresholds and approximately 10 percent of the weights are sent to zero. (b) Angle distribution between continuous and ternary weight vectors by layer. The vertical dotted line (at $\theta \approx 34°$) indicates the theoretical prediction given the empirical ternarization threshold ($\approx 0.11$). $d$ is the dimension of the filters at each layer. (c) Distribution of angles between two random vectors (blue), and between a vector and its quantized version (red), for a rotationally invariant distribution of dimension $d$. The ternarization threshold is chosen to match the trained network. As the red and blue curves have little overlap, ternarization causes a small change in angle in high dimensions. (d) Angle between a random vector and its ternarized version as a function of ternarization threshold for $d = 1000$. There is a large variation in the angle over different thresholds. (e) Ternarization preserves dot products. The 2D histogram shows the dot products between the ternarized weights and the activations (horizontal axis) and the dot products between the continuous weights and the activations (vertical axis) for layer 3 of the network. The dot products are highly correlated.

