# OpenReview forum: "The High-Dimensional Geometry of Binary Neural Networks"
_ICLR.cc/2018/Conference — Accept (Poster)_

### Official Review · AnonReviewer1 · 2017-11-25
**Interesting, yet partially confusing**

**Rating:** 7
**Confidence:** 4

**Review:**

This paper investigates numerically and theoretically the reasons behind the empirical success of binarized neural networks. Specifically, they observe that:

(1) The angle between continuous vectors sampled from a spherical symmetric distribution and their binarized version is relatively small in high dimensions (proven to be about 37 degrees when the dimension goes to infinity), and this demonstrated empirically to be true for the binarized weight matrices of a convenet.

(2) Except the first layer, the dot product of weights*activations in each layer is highly correlated with the dot product of (binarized weights)*activations in each layer. There is also a strong correlation between (binarized weights)*activations and (binarized weights)*(binarized activations). This is claimed to entail that the continuous weights of the binarized neural net approximate the continuous weights of a non-binarized neural net trained in the same manner.

(3) To correct the issue with the first layer in (2) it is suggested to use a random rotation, or simply use continues weights in that layer.

The first observation is interesting, is explained clearly and convincingly, and is novel to the best of my knowledge.

The second observation is much less clear to me. Specifically,
a.	The author claim that “A sufficient condition for \delta u to be the same in both cases is L’(x = f(u)) ~ L’(x = g(u))”. However, I’m not sure if I see why this is true: in a binarized neural net, u also changes, since the previous layers are also binarized.
b.	Related to the previous issue, it is not clear to me if in figure 3 and 5, did the authors binarize the activations of that specific layer or all the layers? If it is the first case, I would be interested to know the latter: It is possible that if all layers are binarized, then the differences between the binarized and non-binarized version become more amplified.
c.	For BNNs, where both the weights and activations are binarized, shouldn’t we compare weights*activations to (binarized weights)*(binarized activations)?
d.	To make sure, in figure 4, the permutation of the activations was randomized (independently) for each data sample? If not, then C is not proportional the identity matrix, as claimed in section 5.3.
e.	It is not completely clear to me that batch-normalization takes care of the scale constant (if so, then why did XNOR-NET needed an additional scale constant?), perhaps this should be further clarified.

The third observation seems less useful to me. Though a random rotation may improve angle preservation in certain cases (as demonstrated in Figure 4), it may hurt classification performance (e.g., distinguishing between 6 and 9 in MNIST). Furthermore, since it uses non-binary operations, it is not clear if this rotation may have some benefits (in terms of resource efficiency) over simply keeping the input layer non-binarized.

To summarize, the first part is interesting and nice, the second part was not clear to me, and the last part does not seem very useful.

%%% After Author's response %%%
a. My mistake. Perhaps it should be clarified in the text that u are the weights. I thought that g(u) is a forward propagation function, and therefore u is the neural input (i.e., pre-activation).

Following the author's response and revisions, I have raised my grade.

---

> ### Author Response · Authors · 2017-12-16
> **Response to comments, and changes made**
>
> Reviewer 3: thank you for your detailed questions and suggestions for our paper.
>
> a. In this argument, u is the weights, and f, g are the identity function / the pointwise binarize function. The earlier layers don’t impact the weights.  I’m not sure I understand your comment.
> b. During training, all of the weights and activations are binarized [except the first layer activations which are the input, and the last layer weights which interface with the output].  We can extract out the values prior to binarization.  In this sense, there isn’t any accumulation of errors.  In other words, Figs 3 and 5 don’t reflect any accumulation of errors.
> c. The reason for only changing one of the binarization of the weights or activations is that corresponds to removing one of the binarize blocks.  However, for the sake of completeness, I included this figure in the SI as well.
> d. The point of the permutation was to generate a distribution with the same marginal statistics, but with no correlational structure.  Each set of activations were independently permuted [clarified in the paper].
> e. Batch normalization subtracts the mean and divides by the standard deviation, then multiplies by a learnable constant and adds a learnable constant.  So there is implicitly a learnable constant multiplying the result of the dot products. However, empirically, the learnable additive constant is zero, so the multiplicative constant is not necessary for all but the last layer at test time because the output of the batch norm layer is subsequently binarized with a threshold of zero.
>
> As far as your question about rotation being a problem with MNIST, the suggestion of our paper is to apply a random rotation to the image as if it is a vector, not to rotate in image space. [Rotation in image space wouldn’t fix the problem because it preserves the correlations between neighboring pixels]. It is the same random rotation for all inputs [added a word to make this more clear].  In the case of MNIST, this is akin to the fact that most of the variance in the dataset happens in the middle of the image and there is almost no variance in the pixels on the edge. The rotation spreads the variance more evenly among the pixels. Of course one potential problem is that the convolutional structure is broken.
>
> One other point as far as the last section is concerned, a number of papers have reported difficulties with the first layer, and we are the first (to my knowledge) to connect this issue to correlations in the data reducing the effective dimensionality of the input. Maybe it isn’t the most ground breaking point, but it seemed worth including to me.
>
> Thanks again for your detailed comments.

---

### Official Review · AnonReviewer3 · 2017-11-26
**The results are not useful**

**Rating:** 4
**Confidence:** 3

**Review:**

This paper presents three observations to understand binary network in Courbariaux, Hubara et al. (2016). My main concerns are on the usage of the given observations.

1. Can the observations be used to explain more recent works?

Indeed, Courbariaux, Hubara et al. (2016) is a good and pioneered work on the binary network. However, as the authors mentioned, there are more recent works which give better performance than this one. For example, we can use +1, 0, -1 to approximate the weights. Besides, [a] has also shown a carefully designed post-processing binary network can already give very good performance. So, how can the given observations be used to explain more recent works?

2. How can the given observations be used to improve Courbariaux, Hubara et al. (2016)?

The authors call their findings theory. From this perspective, I wish to see more mathematical analysis rather than just doing experiments and showing some interesting observations. Besides, giving interesting observations is not good enough. I wish to see how they can be used to improve binary networks.

Reference
[a]. Network sketching: exploiting binary structure in deep CNNs. CVPR 2017

---

> ### Author Response · Authors · 2017-12-16
> **Response to comments**
>
> Reviewer 2: thank you for your consideration of our paper.
>
> 1. I agree that our observations are relevant to understanding the work that seeks to use a ternary representation instead of a binary one [or a higher order quantization].  I’m working on some additional experiments, but it requires a substantial amount of work so that will be included in the next revision if I can get it done in time. Thanks for your pointer to the network sketching paper - I think that our work has interesting connections to it.  However, an analysis of this paper is outside the scope of this work.
> 2. The goal of this paper is to explain why the Courbariaux paper worked as well as it did. I agree that it would be an interesting research direction to improve their work.
>
> As far as your comment that the paper just does some experiments and presents observations, as the other reviewers have noted, the dotted lines in Fig 2b and 2c are theoretical predictions based on assuming a rotationally invariant distribution for the weights, [the proofs are in the SI], and the colored curves/points are the experimental results.  There is a close correspondence between the theory and the experiments.
>
> More broadly, I agree that it would be great if this analysis led to a technique for improving performance of binary neural networks. However, I believe that the results already paint an insightful picture for understanding binary neural networks that would be useful to share with the community.  Regardless, thank you for your suggestions for further experiments, hopefully I can improve the paper to your liking.

---

> > ### Author Response · Authors · 2017-12-22
> > **Analysis of Ternary Neural Networks added**
> >
> > Reviewer 2: Pursuant of your suggestions, I added a section that uses our approach to analyze ternary neural networks.  The methods that were developed for binary neural networks generalize nicely to TNNs [although it is slightly more complicated as TNNs have an extra parameter in the quantization function]!  Thanks for your push to address this subject.   The new analysis is included in the latest draft of the paper.   If you have other questions, comments, or suggestions, let me know!

---

### Official Review · AnonReviewer2 · 2017-11-27
**An insightful analysis on binary neural networks and its learning dynamics**

**Rating:** 7
**Confidence:** 4

**Review:**

This paper tries to analyze the effectiveness of binary nets from a perspective originated from the angular perturbation that binarization process brings to the original weight vector. It further explains why binarization is able to preserve the model performance by analyzing the weight-activation dot product with "Dot Product Proportionality Property." It also proposes "Generalized Binarization Transformation" for the first layer of a neural network.

In general, I think the paper is written clearly and in detail. Some typos and minor issues are listed in the "Cons" part below.

Pros:
The authors lead a very nice exploration into the binary nets in the paper, from the most basic analysis on the converging angle between original and binarized weight vectors, to how this convergence could affect the weight-activation dot product, to pointing out that binarization affects differently on the first layer. Many empirical and theoretical proofs are given, as well as some practical tricks that could be useful for diagnosing binary nets in the future.

Cons:
* it seems that there are quite some typos in the paper, for example:
    1. Section 1, in the second contribution, there are two "then"s.
    2. Section 1, the citation format of "Bengio et al. (2013)" should be "(Bengio et al. 2013)".
* Section 2, there is an ordering mistake in introducing Han et al.'s work, DeepComporession actually comes before the DSD.
* Fig 2(c), the correlation between the theoretical expectation and angle distribution from (b) seems not very clear.
* In appendix, Section 5.1, Lemma 1. Could you include some of the steps in getting g(\row) to make it clearer? I think the length of the proof won't matter a lot since it is already in the appendix, but it makes the reader a lot easier to understand it.

---

> ### Author Response · Authors · 2017-12-16
> **Changes made**
>
> Reviewer 1: thank you very much for your comments.
>
> * Typos fixed.
> * Ordering of the papers:  the Han Deep Compression paper cites the Han Learning weights paper. [So that order is correct].  However, a citation to the DSD paper, which is more recent than those two earlier papers, is missing.  I added a citation this to the paper.
> * Fig 2(b) shows the angle distributions separated by layer [and each layer has different dimensional vectors].  Each of these distributions is peaked with some standard deviation.  The theory predicts that these distributions have standard deviation that scales as 1/sqrt(d).  The plot in 2(c) shows the standard deviation as a function of dimension of the curves in 2(b) with a dotted line that corresponds to 1/sqrt(d)  [on a log-log plot].  The dots fall roughly along this dotted line (especially for higher dimensions).  For the sake of clarity, I added the zoomed in version of the plot in 2(b) in the appendix. If you have any suggestions for how to make this more clear, I am happy to fix it.
> *  The proof for the pdf of g(rho) is a bit involved in the paper that I cited. I came up with a new proof that is quite a bit simpler and included it.

---

### Public Comment · (anonymous) · 2017-12-21
**Histogram of weight distribution**

The result shown in figure 6d is actually quite different from what we observed in experiments. Distribution of weights are actually highly dependent on the learning rate being used.

---

> ### Author Response · Authors · 2017-12-21
> **We decay the learning rate**
>
> Hello, thanks for your interest in this work!  We follow the method of the Courbariaux paper and use ADAM while simultaneously decaying the learning rate from 0.001 to 0.0000003 over 500 epochs of training.  I am not sure what analysis you are using.  I think that any optimization method that is looking at the continuous weights should decay the learning rate. The discrete nature of the forward function pass of the binarize function sets a scale for the size of the subsequent gradients going backward.  This fixed size needs to be scaled down by a reduced learning rate to get to the best values of the continuous weights.

---

> > ### Public Comment · (anonymous) · 2018-01-09
> > **Your theory depend on a hyper-parameter of the model**
> >
> > If the evidence your theory depends on can go away under different learning rate (a hyper-parameter) setting, I don't believe your theory can still stand.
> >
> > Or we can put it in another way , let's scale the continuous weights by a fixed number ( for example, 3.0), and we keep the binarization part same. In this way we still have same {-1,+1} weights, and the network will still work (may require some adjustment in learning rate). Clearly we can no longer claim the binarized weights are approximation of the continuous weights here.
> >
> > Hope my input can help us gain better understanding of binarized neural network.

---

> > > ### Author Response · Authors · 2018-01-09
> > > **Weight scaling taken into account in the analysis**
> > >
> > > As far as your second comment is concerned, the possibility of scaling the continuous weights is already taken into account in the theory. First, the angle preservation property looks at the angle between the continuous weight and the binary weight, which is independent of the scaling of the continuous weight.  Second, the dot product *proportionality* property notes that the continuous weight activation dot product is proportional to the binary weight activation dot product.  If the continuous weights are scaled by a factor of two, that just changes this proportionality constant.
> > >
> > > Just to reiterate my point from my previous comment, keeping the learning rate fixed isn't running the network to convergence, so it doesn't make sense to have a fixed learning rate.
> > >
> > > If you would like to pursue the weight distribution point further, you should demonstrate that your networks converge when you use a fixed learning rate (in particular, train the network with a fixed learning rate, save the parameters, and then reduce the learning rate, and show that the continuous weights don't change after running learning with the reduced learning rate, which I think is unlikely to be the case).   Just as another important point, the network performance can stabilize while the weights are still not converged.  This is a subtle point because small fluctuations around the weights may not change dot products substantially, but when the training is run for longer, these small fluctuations go away.
> > >
> > >
> > > Furthermore, it isn't clear to me that you can just arbitrarily scale the weights without changing the learning algorithm.  This is because there are two scale parameters associated with the weights. First, after each gradient step update, the continuous weights are clipped to be between -1 and +1.  So you could scale this clipping threshold.  However, there is also the parameter where the backwards pass of the binarization function clips the gradient at -1 and +1 as well (  g(x) = x for |x|< 1, and sign(x) for |x|>=1).  This function can't just be scaled in the same way because g(x) needs to match the forward function for large x, and f(x), the binarization function outputs +1 and -1. Scaling the threshold for g(x) breaks the relationship between f and g.
> > >
> > > Let me know what you think.

---

### Decision · Program_Chairs · 2018-01-29
**ICLR 2018 Conference Acceptance Decision**

**Decision:**

Accept (Poster)

**Comment:**

This paper analyzes mathematically why weights of trained networks can be replaced with ternary weights without much loss in accuracy. Understanding this is an important problem, as binary or ternary weights can be much more efficient on limited hardware, and we've seen much empirical success of binarization schemes. This paper shows that the continuous angles and dot products are well approximated in the discretized network. The paper concludes with an input rotation trick to fix discretization failures in the first layer.

Overall, the contribution seems substantial, and the reviewers haven't found any significant issues. One reviewer wasn't convinced of the problem's importance, but I disagree here. I think the paper will plausibly be helpful for guiding architectural and algorithmic decisions. I recommend acceptance.